# The Effects of Childhood Maltreatment on Non-Suicidal Self-Injury in Male Adolescents: The Moderating Roles of the Monoamine Oxidase A (MAOA) Gene and the Catechol-O-Methyltransferase (COMT) Gene

**DOI:** 10.3390/ijerph18052598

**Published:** 2021-03-05

**Authors:** Yemiao Gao, Yuke Xiong, Xia Liu, Hui Wang

**Affiliations:** Institute of Developmental Psychology, Beijing Normal University, Beijing 100875, China; bnuymgao@mail.bnu.edu.cn (Y.G.); 202031630011@mail.bnu.edu.cn (Y.X.); xiaohui.wang@mail.bnu.edu.cn (H.W.)

**Keywords:** childhood maltreatment, MAOA gene, COMT gene, adolescent NSSI, gene–gene–environment interaction

## Abstract

(1) Background: Numerous studies suggest strong associations between childhood maltreatment and nonsuicidal self-injury (NSSI); this is also true for the roles of dopaminergic genes in the etiology of some psychopathologies related to NSSI. Investigating the interactions of environments and genes is important in order to better understand the etiology of NSSI. (2) Methods: Within a sample of 269 Chinese male adolescents (*M*age = 14.72, *SD* = 0.92), childhood maltreatment and NSSI were evaluated, and saliva samples were collected for MAOA T941G and COMT Val158Met polymorphism analyses. (3) Results: The results revealed no primary effects attributable to MAOA T941G and COMT Val158Met polymorphism on NSSI. However, there was a significant three-way interaction between MAOA, COMT, and child abuse (β = −0.34, *p* < 0.01) in adolescent NSSI. Except for carriers of the T allele of MAOA and the Met allele of COMT, all studied male adolescents displayed higher NSSI scores when exposed to a higher level of child abuse. A similar three-way interaction was not observed in the case of child neglect. (4) Conclusions: The results indicate that the MAOA gene and COMT gene play moderating roles in the association between child abuse and NSSI of male adolescents and suggest the polygenic underpinnings of NSSI.

## 1. Introduction

Nonsuicidal self-injurious (NSSI) behavior—defined as “the direct, deliberate destruction of one’s body tissue in the absence of intent to die” [1]—is a significant mental health concern. The onset of NSSI typically occurs in early adolescents, with an approximate age between 12 and 14 years old [2]. Research with community samples shows that the prevalence of adolescent NSSI ranges from 17.2–38.6% across the world [3,4]. Although this kind of behavior is usually nonfatal, individuals engaged in NSSI are at greater risk for suicide [5,6]. Thus, identifying the associated factors that contribute to the etiology of NSSI is important for developing effective prevention and intervention strategies for this behavior. Theoretical models and empirical evidence of NSSI emphasize the roles of both environmental and biological factors in the development of this behavior [1,7]. In recent years, a burgeoning body of research has examined the etiology of various kinds of psychiatric disorders through the perspective of gene–environment interactions [8,9]; however, limited research on NSSI from this perspective limits the understanding of the development of this behavior. Thus, in the current study, we investigated childhood maltreatment (i.e., abuse and neglect) and two candidate genes (i.e., MAOA T941G and COMT Val158Met) to examine their main and interactive effects on NSSI, with a sample of Chinese male adolescents.

### 1.1. Childhood Maltreatment and NSSI

Childhood maltreatment (i.e., abuse and neglect) has been found to be a particularly salient risk factor for NSSI [10]. According to the developmental psychopathology framework of self-injurious behavior [11], negative experiences in early childhood, particularly in the caregiving environment, could lead to disruptions in adaptive skill development. The vulnerabilities in adaptive functioning can in turn contribute to the emergence of maladaptive coping strategies—such as NSSI. Nock’s theoretical model of NSSI [1] also posits that child maltreatment, as a distal environmental factor, may increase the risk of problems with emotional regulation and interpersonal communication—which are vulnerability factors related to NSSI. Indeed, empirical research has provided extensive supportive evidence for the associations between childhood maltreatment and NSSI [12,13,14,15]. 

However, adolescents exposed to childhood maltreatment vary widely in their vulnerability to NSSI. In terms of the associations between specific subtypes of childhood maltreatment and NSSI, the existing literature has also shown mixed findings [12,13,14,15]. For example, one systematic review revealed that sexual abuse was more related to NSSI than other kinds of maltreatment [16], while another systematic review found that emotional abuse had the highest degree of association [10]. One hypothesis to explain these discrepant results is that the impact of invalidating environments might be moderated by other factors. Amid the numerous factors used to explain variabilities in the expressions of problem behaviors, biological factors—such as genes—have gained increasing attention [17].

### 1.2. Dopaminergic Genes and NSSI

The four-function model of nonsuicidal self-injury and the experiential avoidance model have both demonstrated that one of the functions of NSSI behavior is to regulate affective states immediately [1,18]. It seems appropriate to hypothesize that brain functions underpinning emotional regulation or impulsiveness are associated with NSSI. Therefore, dopaminergic genes may be fruitful candidates for NSSI vulnerability. 

The monoamine oxidase A (MAOA) gene, which is located on the X chromosome (Xp11.23–11.4), is one of the dopaminergic genes. It encodes an enzyme that is responsible for the degradation of monoamine neurotransmitters in the brain, including norepinephrine (NE), dopamine (DA), and serotonin (5-HT). The monoamine deficiency hypothesis of depression [19] posits that a deficiency in serotonin or norepinephrine neurotransmission in the brain plays a role in depressive symptoms—which are common in individuals who engage in NSSI. Furthermore, it has been suggested that dysregulation of monoamine neurotransmissions has an important effect on brain circuits, one concerned with emotional regulation and responses to psychological stress [20]. Previous research has provided sufficient support to suggest that MAOA gene polymorphism has an impact on depression and impulsivity (which are closely associated with self-injurious behaviors) via the development and functioning of corticolimbic circuits [21,22]. Thus, adolescents are likely to show different sensitivity to adverse life experiences due to the genotype variability of MAOA [23]. 

The T941G polymorphism in exon 8 is a common functional polymorphism of the MAOA gene. A substitution of T in place of G at this location could result in a 75% improvement in MAOA activity [24]. Broadly, the literature has investigated the moderating role of MAOA gene polymorphism in people’s response to childhood maltreatment—albeit with inconsistent results. Some studies showed that high activity alleles of MAOA genes conferred risk for depressive symptoms [9] or borderline personality disorder symptoms [25] in the context of childhood maltreatment. Others found that the interaction of low MAOA activity and child maltreatment predicted higher depressive symptoms [26] or greater impulsiveness [27]. These mixed findings have resulted in skepticism regarding the use of single locus in MAOA genes when examining gene-environment interactions. 

The catechol-O-methyltransferase (COMT) gene is another well-known dopaminergic system gene. It lies on chromosome 22q11 and encodes the COMT enzyme, which is a major enzyme involved in the degradation of dopamine, norepinephrine, and epinephrine [28]. The COMT gene contains a functional single nucleotide polymorphism (SNP), rs4680, which produces a valine (Val) to methionine (Met) substitution at codon 158 (Val158Met); COMT enzyme activity with Met is one-fourth lower than that with Val [29]. The decreased COMT enzyme activity results in higher cortical dopamine levels [30], and therefore inflexible processing of affective stimuli, which is a mechanism that possibly accounts for emotional dysregulation [31]. Since one of the main characteristics of self-injurers is emotional dysregulation, we could propose that the COMT gene polymorphism might also play a role in NSSI engagement.

### 1.3. Gene × Gene × Environment Interactions

The biosocial model of borderline personality [32] designates genes as sources of individual differences in the risk of developing self-injurious behavior following early family adversity. According to the biosocial model, both biological vulnerabilities and adverse caregiving environments increase people’s risk for emotion dysregulation and more extreme behavioral dyscontrol, thereby contributing to impulsive behaviors (including self-injury). Furthermore, the modified diathesis-stress model proposed by Brodsky [17] suggests that childhood adversity interacts with neurobiological and genetic factors to contribute to NSSI vulnerability—not only through certain character traits (e.g., aggression, emotional dysregulation), but also through the biological impact on genetic phenotypes and neurotransmitter systems, including serotonin, opioid, oxytocin, and the hypothalamic-pituitary-adrenal axis. 

Recent research supported the proposition that both genetic and environmental factors contribute to NSSI vulnerability [7]. However, to our knowledge, only two empirical studies have examined the interaction between genes and environmental factors in predicting NSSI. One study by Bresin et al. [33] implicated that the association between childhood emotional environment and NSSI was moderated by BDNF Val66Met polymorphism. Another study investigating the impact of interpersonal stress on NSSI found a significant moderating role of 5-HTTLPR gene polymorphism [34]. Although they represent an advance in understanding the moderating role of genes in the relationship between environments and NSSI, both studies only examined the rate of NSSI engagement; arriving at a better understanding of the effect of gene–environment interaction on the frequency and severity of this behavior is also important. Additionally, both studies focused on a single locus of genes. New molecular genetic evidence has shown that many genetic variants may work together (e.g., genes and alleles interact with each other) in predicting psychological outcomes, but may not act in isolation. Therefore, findings should be interpreted with caution until further replicated. 

Biological evidence indicates that MAO and COMT activities interact to affect dopamine neurotransmission [35] and HPA axis stress response [36]. Empirical evidence also implies that MAOA and COMT genes act interactively to affect NSSI-related traits and other forms of psychopathology [37,38]. In spite of the previous findings that both MAOA and COMT genes are involved in depression [9] and emotional regulation [39]—which are closely related to NSSI—a dearth of studies has examined the roles that these genes play in the association between environmental factors and adolescent NSSI. Additionally, although the relationships between different kinds of maltreatment and NSSI remain unclear, it seems that NSSI has stronger associations with child abuse than with child neglect [10,16]. Therefore, the current study was designed to address these gaps by examining possible gene–gene–environment interactions implicated in the etiology of NSSI behaviors. This will provide new insight into the relationship between childhood maltreatment subtypes and NSSI, as well as neurobiological mechanisms in the development of NSSI.

### 1.4. Current Study

The present study aimed to explore the independent and interactive effects of MAOA gene, COMT gene, and childhood maltreatment (i.e., child abuse and neglect) on adolescent NSSI. The participants in the current study were limited to male adolescents because MAOA is an X-linked gene (whether there is X-inactivation is still controversial) [40]. We expected the MAOA gene and COMT gene to interact with each other in moderating the effects of childhood abuse and neglect on adolescent NSSI behaviors. Specifically, we hypothesized that carriers of the MAOA G allele and COMT Met allele would suffer from high childhood abuse or neglect and hence engage in higher levels of NSSI, while individuals with the MAOA T allele and COMT Val homozygote would not exhibit the same relationship between childhood abuse/neglect and NSSI.

## 2. Materials and Methods

### 2.1. Participants and Procedures

Data for the present study were drawn from a longitudinal project on children and adolescent adjustment. Collaborating with public welfare organizations and a local education center, the project has run for three consecutive years. Four public junior high schools in Guizhou province, China, were selected randomly and each approved the survey. All students from these schools were Chinese. In the current study, we used the data from the last wave of the project. Random cluster sampling was used to choose 14 classes in grade nine across the four schools. A total of 570 consenting participants completed the survey in the larger study. Inclusion criteria consisted of adolescents who had no major diseases and functioning intellectual capacity (so that they were able to complete study questionnaires). Informed assent forms from participants (as well as signed consent forms from their parents and school principals) were obtained before data collection. 

Since MAOA is an X-linked gene, it is still controversial whether there is X-inactivation [40]. In heterozygous females, it is difficult to determine the effect of each allele. Therefore, based on existing studies [23,28], we excluded female cases from the analysis. Participants in the present study consisted of 269 male students (mean age = 14.72, *SD* = 0.92). Parents had a median educational level equivalent to senior high school. More than half (57.6%, *n* = 155) of the families had monthly household incomes below 4000 yuan ($565); 33% (*n* = 89) had incomes of 4000–8000 yuan ($565–$1130); and 6.3% (*n* = 17) had incomes higher than 8000 yuan ($1130). 

Participants were invited to complete self-reported questionnaires and provide saliva samples for DNA extraction, both of which were performed in the classroom. During data collection, there was at least one trained research assistant in each classroom, providing professional help and checking saliva samples. Each participant received a gift after data collection. The Research Ethics Committee of the authors’ university approved the study.

### 2.2. Measures

#### 2.2.1. Child Abuse and Neglect

Child abuse and neglect were measured using the translated version of the Childhood Trauma Questionnaire-Short Form (CTQ-SF [41]). The CTQ-SF is a 28-item retrospective measure consisting of three dimensions of child abuse and two dimensions of neglect: emotional, physical, and sexual abuse, and emotional and physical neglect. Participants rated the frequency (ranging from 1 = never to 5 = very often) of each item. Responses of the corresponding subscales were summed, with a high score representing a high level of child abuse or neglect. The Chinese version of the CTQ-SF showed adequate internal consistency [42,43]. To evaluate the prevalence of childhood maltreatment, we classified a type of childhood trauma as existing moderate-severe maltreatment if the subscale score was higher than the cut-off points suggested by other studies [43,44]: emotional abuse ≥ 13, physical abuse ≥ 10, sexual abuse ≥ 8, emotional neglect ≥ 15, physical neglect ≥ 10. The Cronbach’s α coefficient for the total scale was 0.86.

#### 2.2.2. Nonsuicidal Self-Injury (NSSI)

Participants completed the modified version of the Adolescents Self-Harm Scale (DSHS) [45] to assess their self-injurious behaviors without suicidal intent. The DSHS contains 19 items, with a checklist of 18 NSSI behaviors (e.g., self-cutting with glass or a knife) including frequency and severity, and an open question measuring NSSI behavior that is not listed in the questionnaire. Participants rated each of the listed behaviors on a 4-point scale (1 = never, 2 = once, 3 = twice to four times, and 4 = five times or more) to assess lifetime NSSI frequency. Participants rated each of the listed behaviors on a 5-point scale (ranging from 1 = no damage to 5 = very severe) to evaluate the degree of physical damage caused by the behavior. The score of the NSSI was calculated by summing the products of frequency and severity scores. Higher DSHS scores indicated greater self-injurious experiences. The DSHS has demonstrated good validity and reliability among Chinese adolescent populations [46,47]. Cronbach’s α for the DSHS in the present study was 0.93.

#### 2.2.3. Covariables

Evidence suggests that depression is associated with NSSI (e.g., [48,49]), so we regarded depressive symptoms as a covariate in the subsequent analysis. Depressive symptoms were measured using the Center for Epidemiologic Studies Depression Scale for Children (CES-DC) [50]. Higher scores indicated higher levels of depressive symptoms. The Cronbach’s α was 0.87 in the present study. Age was also specified as a covariate because of the strong correlation between NSSI prevalence and age [48].

#### 2.2.4. Genotyping

Saliva samples were collected from participants for genomic DNA extraction. Genotyping was performed through matrix-assisted laser desorption ionization time-of-flight mass spectrometry method (MALDI-TOF) in the MassARRAY system (Sequenom Inc., San Diego, CA, USA) following the manufacturer’s recommendations. Genotyping of the MAOA gene T941G polymorphism was carried out with forward primer sequences ACGTTGGATGTGCACTTAAATGACAGTCCC and reverse primer sequences ACGTTGGATGGATTCACTTCAGACCAGAGC. The COMT gene Val158Met polymorphism was amplified using the forward primer sequence ACGTTGGATGACCCAGCGGATGGTGGATTT and reverse primer sequence ACGTTGGATGTTTTCCAGGTCTGACAACGG. 

### 2.3. Data Analyses

Bivariate Pearson correlations were examined to evaluate the associations between main variables in the present study and to determine whether environmental factors were correlated with genes. A set of hierarchical linear regression analyses were conducted to test the main and interactive effects of MAOA gene, COMT gene, and childhood maltreatment on NSSI. Child abuse and neglect were tested in separate models. All continuous predictive variables were mean-centered before analyses to aid the interpretation of interactive effects. Since NSSI did not meet assumptions of normality, we used log transformation to improve the normality of the NSSI variable first. However, additional regression analyses with raw data produced almost identical result patterns to the analysis with transformed data. Thus, we retained and reported here the analyses of the original variable for meaningful interpretation of results. Missing data were listwise deleted. Additionally, to control Type-I error, we corrected the corresponding *p* level through the Benjamini–Hochberg procedure [51]. Simple slope analyses were performed to further investigate significant gene–gene–environment interactions. All the above analyses were conducted using SPSS 26.0 software (IBM Corp., Armonk, NY, USA).

## 3. Results

### 3.1. Hardy-Weinberg Equilibrium Test

There were 35 missing DNA samples (13.01%) due to failure in the collection of saliva samples or other reasons. Results of independent-sample *t*-tests showed that there were no significant differences in child abuse (*t* = −0.77, *p* > 0.05), neglect (*t* = −0.60, *p* > 0.05), or NSSI (*t* = −0.14, *p* > 0.05) between adolescents with and without DNA samples. Therefore, we contained the statistics of all participants.

Distributions of the genetic polymorphisms were as follows: MAOA G = 44.98% (*n* = 121), T = 41.26% (*n* = 111); COMT Met/Met = 4.83% (*n* = 13), Val/Met = 33.09% (*n* = 89), Val/Val = 49.07% (*n* = 132); the polymorphism was in Hardy–Weinberg equilibrium (χ^2^ = 0.16, *df* = 2, *p* > 0.05). Consistent with previous research [52,53], we collapsed Met/Met with Val/Met genotype groups and dummy-coded the MAOA and COMT genotypes into 1 = G allele or Val/Val genotype and 0 = T allele or presence of Met allele (i.e., Met/Met and Val/Met).

### 3.2. Descriptive Statistics and Correlations

A total of 51.2% participants reported having engaged in at least one type of NSSI. The most common self-injury methods reported were hitting (37.9%; *n* = 102), pulling hair (37.2%; *n* = 100), and cutting (32%; *n* = 86). 38.3% (*n* = 103) of the sample reported having experienced at least one kind of maltreatment in childhood: 19 subjects (7.1%) reported emotional abuse, 37 subjects (13.8%) reported physical abuse, 31 subjects (11.5%) reported sexual abuse, 46 subjects (17.1%) reported emotional neglect, and 54 subjects (20.1%) reported physical neglect.

Table 1 shows the means, standard deviations, and bivariate correlations among the main variables. MAOA gene and COMT gene were uncorrelated with child abuse and neglect, indicating the absence of association between genes and environment. Both child abuse (*r* = 0.42, *p* < 0.001) and neglect (*r* = 0.18, *p* < 0.01) were positively associated with NSSI.

### 3.3. Interaction Effects of MAOA, COMT, and Child Abuse on NSSI

Table 2 presents the results of the regression analysis testing the moderating effects of MAOA gene and COMT gene on the relationship between child abuse/neglect and NSSI, after controlling for the effects of age and depressive symptoms. The power to detect the significant effect (*p* < 0.05) with the sample size in the study was more than 99%, under an effect size index of 0.30.

As for child abuse, MAOA and COMT were not significantly associated with NSSI, but both served as moderators for the effects of child abuse on NSSI (β = 0.35, *t* = 3.00, and β = 0.42, *t* = 3.71 respectively, *p*s < 0.01). There was also a significant three-way interaction among the MAOA genotype, COMT genotype and child abuse (β = −0.34, *t* = −3.22, *p* < 0.01). To further interpret the three-way interaction effect, we conducted a simple slope analysis. As shown in Figure 1, there was not significant association between child abuse and NSSI in Met/T genotypes carriers (β = 0.12, *t* = 0.59, *p* > 0.05), while in carriers of the other combination genotypes, the score of child abuse positively predicted NSSI (Met/G carriers: β = 0.81, *t* = 7.53, *p* < 0.001; Val/Val/T: β = 0.65, *t* = 5.70, *p* < 0.001; Val/Val/G: β = 0.28, *t* = 2.14, *p* < 0.05).

There was no significant main effect (|β|s ≤ 0.13, |*t*|s ≤ 1.10, *p*s > 0.05) nor significant two-way (|β|s ≤ 0.21, |*t*|s ≤ 1.64, *p*s > 0.05) or three-way interactions (β = −0.18, *t* = −1.34, *p* > 0.05) among MAOA, COMT and child neglect on adolescent boys’ NSSI. 

## 4. Discussion

Previous studies have shed light on the importance of gene–environment interactions in furthering our understanding of the etiology of NSSI [33,34]. However, accumulating evidence suggests that many genetic variants may not act in isolation in predicting psychological outcomes [54,55]. Therefore, from a gene–gene–environment interaction perspective, the present study investigated the individual and joint effects of MAOA T941G and COMT Val158Met polymorphisms, two important dopamine system genes, and childhood maltreatment on male adolescents’ NSSI behavior. Our findings suggested that neither MAOA T941G nor COMT Val158Met polymorphism had a direct impact on NSSI; however, a significant three-way interaction effect was found among MAOA, COMT, and child abuse in relation to NSSI.

No direct associations between the MAOA or the COMT genotypes and NSSI emerged in the present study, indicating that MAOA and COMT genes did not directly place male adolescents at heightened risk for NSSI. The results were in accord with other studies revealing that the effects of single polymorphism on NSSI and NSSI-related traits (e.g., depression) were always small or negligible [33,56]. Additionally, our results did not show the main effects of child abuse or neglect on NSSI, contrary to some studies [57,58,59]. Recent reviews have suggested that the diversity of participants (e.g., gender difference), the existence of mediators [16], or the type of maltreatment [60] could contribute to heterogeneity among studies. For example, in the present study, the score of depression showed significant correlations with child abuse and neglect as well as NSSI. Combined with previous literature, the results indicated that depressive symptoms might play a mediating role between childhood maltreatment and NSSI [60]. Furthermore, we only tested the relationship between childhood maltreatment and NSSI in a sample of male adolescents and used the combination of specific types of abuse or neglect, which might explain the discrepancy between the current and previous studies. 

With regard to the susceptibility of the MAOA gene, some theoretical and empirical studies have suggested that the G allele of the MAOA gene may be the susceptible allele to adverse environments [9,19], while other studies reached the opposite conclusion [26,27]. The heterogeneous outcomes might result from potential polygenetic effects and suggest the importance of taking multiple gene–environment interaction perspectives. In this study, we found significant three-way interaction effects between child abuse, the MAOA gene, and the COMT gene. Specifically, the G allele of the MAOA gene exerted its negative impact on NSSI after increased child abuse in the presence of all COMT genotypes. Additionally, the T allele of the MAOA emerged as a genetic susceptibility factor for NSSI in male carriers of the COMT Val/Val genotype. This interaction effect provided a better understanding of the polygenic underpinnings of NSSI, as well as the interactive effects between the MAOA gene and adverse childhood experiences. 

Previous studies have shown that higher MAOA activity, which results in lower dopamine levels, is associated with reduced corticolimbic connectivity [61] and increased sensitivity to negative emotional stimuli [62]. MAOA G allele carriers’ emotional response to environmental stimuli could therefore be amplified by the neurocircuits critical for emotional regulation. Given that emotional regulation is the main function of NSSI [1], G allele carriers who experience child abuse might have difficulties in regulating negative emotions and be more likely to engage in NSSI. 

Moreover, epistatic effects of the COMT and MAOA genes have been revealed in the current study, demonstrating that the COMT Val/Val genotype potentiated the susceptibility of the MAOA T allele. Existing studies have provided evidence that high activity COMT genotypes (Val/Val) confer heightened susceptibility to negative environments [37,63]. Additionally, researchers have found increased behavioral sensitivity to emotional stimuli in high activity Val allele homozygous participants [64]. It is also demonstrated that the COMT Val/Val genotype was associated with decreased amygdala–prefrontal connectivity [65]. According to the late-maturing prefrontal cortex (PFC) theory, the decreased modulation of the PFC effect on limbic structures might provoke emotional dysregulation, hence rendering adolescents susceptible to their environments [66]. 

Although the mechanisms of the MAOA by the COMT interaction effect remain unclear, it seems that the decreased amygdala–prefrontal connectivity contributed by the COMT Val/Val genotype puts MAOA T carriers with higher exposure to child abuse at risk of engaging in NSSI. Additionally, both the MAOA and COMT genes are related to serotonin and dopamine neurotransmission in the brain, the deficiency of which (in high activity alleles) is assumed to have effects on depressive symptoms [19], hence increasing the risk for NSSI among adolescents with abusive experiences. From the above, possessing the MAOA G allele or COMT Val/Val genotypes may render male adolescents more susceptible to their environments via lower dopaminergic reactivity in the brain, as well as the decreased capacity of the PFC to modulate amygdala activity.

However, there were no significant interaction effects between child neglect, MAOA, and COMT genes. Child neglect refers to the failure to provide for emotional and physical needs, rather than direct, initiative harmful behaviors (e.g., abuse); thus, child neglect may bring less direct and severe harm to children than abuse [67], especially for males [14]. Previous studies have demonstrated that child neglect, relative to abuse, had a smaller or nonsignificant effect on adolescent psychological and behavioral problems such as impulsiveness and aggression [68,69]. Our findings also suggest that child neglect may play a less important role in male adolescents engaging in NSSI behaviors than child abuse [10,70]. However, child neglect has been shown to have stronger effects on other developmental problems in adolescents (e.g., school adjustment) [71], so attention should be paid to adolescents with neglected experiences in order to improve their development.

Of note, our findings did not suggest that high activity alleles of the MAOA and COMT genes were risk factors for self-injury, but rather that they altered adolescents’ susceptibility to rearing environments [72,73]. Although we did not investigate the effects of positive rearing environments’ interaction with genes, a trend did emerge that adolescents carrying high activity alleles of MAOA and COMT genes—relative to adolescents with Met-T genotype—showed less NSSI behavior when exposed to a low level of child abuse. Thus, it would be more appropriate to conceptualize the high activity alleles of MAOA and COMT genes as modifying sensitivity to adverse childhood experiences that confer risk for adolescent NSSI.

For the first time, we investigated the main and interactive effects of childhood maltreatment, the MAOA gene T941G, and the COMT Val158Met polymorphisms in predicting NSSI with a sample of Chinese adolescents. The current study provides new empirical evidence for the role of genes as a modulating mechanism between childhood maltreatment and NSSI, as well as potential implications for providing support for the primary prevention of child maltreatment and adolescent maladaptive behavior. For example, screening male adolescents for child abuse history may be more important than screening for neglect history in the assessment of risk for NSSI. Addtionally, male carriers of the G allele of MAOA T941G or Val/Val genotype of COMT Val158Met might benefit from early intervention in and prevention of NSSI after moderate to severe childhood abuse.

It is also necessary to consider several limitations of the present study. First, a relatively small gene–gene–environment interaction effect was detected in this study, which is common in gene–environment interaction studies, even for outcome variables with high heritability such as depression [74]. Additionally, it might be conceivable that some other factors have influenced the interplay between MAOA, COMT, and childhood maltreatment, such as the serotonergic system, positive environmental factors, and so on. Meanwhile, larger replication studies are needed to confirm our findings. The current study also focused on a rural sample of Chinese males. Future research will need to elucidate whether these results are replicable among adolescents that are representative of different socioeconomic status or racial groups. Third, the current study was cross-sectional and assessed childhood maltreatment and NSSI in retrospect and self-report. In this case, we should keep in mind that there is a possibility of memory bias. The cross-sectional design also limited our ability to reveal the complex interactions or reciprocal relationships between neurobiological system functioning, environmental factors, and behavior [75]. Therefore, examining the relationships between childhood trauma, genes, and NSSI in a dynamic view is an important future direction. Furthermore, although we distinguished child abuse from neglect, previous literature has shown that different types of abuse (i.e., physical/emotional/sexual abuse) may play different roles in NSSI. For example, sexual abuse and emotional abuse have shown stronger associations with NSSI than other kinds of abuse [10,16,76]. Thus, further studies may consider the exact neurobiological mechanisms underlying the effects relating to specific forms of child abuse. 

## 5. Conclusions

To our best knowledge, this is the first study to investigate gene–gene–environment interaction effects on NSSI. The current study provides evidence that two dopaminergic genes modulate Chinese male adolescents’ susceptibility to childhood abuse and supports the multigenetic underlying mechanism of adolescent NSSI. The current results suggest that male adolescents carrying the MAOA G allele, MAOA T allele, or COMT Val/Val genotype are more susceptible to child abuse, relative to the carriers of the combination of MAOA T and COMT Met alleles. Moreover, we did not find significant associations between child neglect and male adolescents’ NSSI. Our findings might also stimulate further studies considering multiple gene–environment interactions in the development of NSSI.

## Figures and Tables

**Figure 1 ijerph-18-02598-f001:**
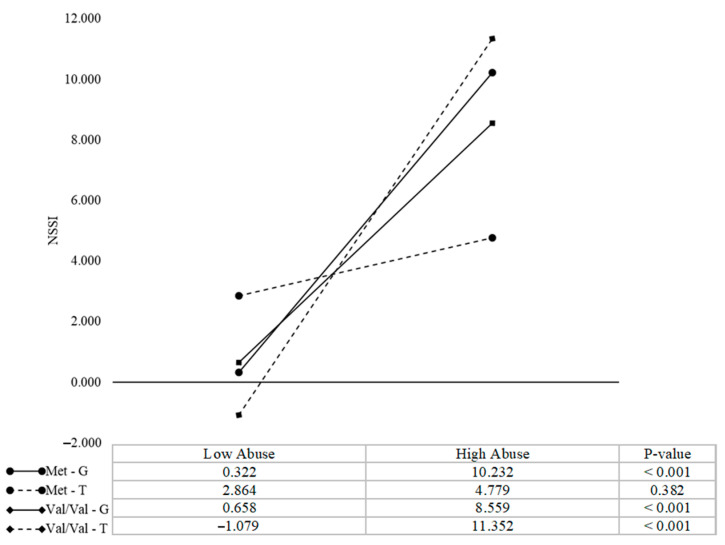
Interaction among MAOA T941G, COMT Val158Met, and child abuse associated with adolescent boys’ nonsuicidal self-injury (NSSI). High and low abuses are defined as +/− one standard deviation from the mean, respectively. All continuous values were mean-centered.

**Table 1 ijerph-18-02598-t001:** Correlations, means, and standard deviations among primary variables.

Variables	1	2	3	4	5	6	7	8	9	10
1. Age	1									
2. Father’s education level	−0.23 ***	1								
3. Mother’s education level	−0.24 ***	0.46 ***	1							
4. Household income	−0.13 *	0.38 ***	0.34 ***	1						
5. Depressive symptoms	0.06	−0.11	−0.10	−0.10	1					
6. MAOA T941G	0.02	−0.00	0.05	−0.10	−0.04	1				
7. COMT Val158Met	−0.16 *	0.06	0.11	0.07	0.04	−0.01	1			
8. Abuse	−0.01	−0.07	−0.02	−0.07	0.47 ***	−0.01	−0.13	1		
9. Neglect	0.16 **	−0.14 *	−0.09	−0.11	0.35 ***	0.08	−0.11	0.42 ***	1	
10. NSSI	0.04	−0.09	−0.03	−0.04	0.38 ***	0.01	−0.01	0.54 ***	0.18 **	1
*M*	14.72	−	−	−	38.71	−	−	20.20	17.73	4.37
*SD*	0.92	−	−	−	9.84	−	−	6.15	6.44	8.66

Note: *N* = 269. NSSI = nonsuicidal self-injury. MAOA = G (1) or T (0); COMT = Val/Val (1) or Met allele (0). * *p* < 0.05; ** *p* < 0.01; *** *p* < 0.001.

**Table 2 ijerph-18-02598-t002:** Male adolescents’ nonsuicidal self-injury (NSSI) regressed on MAOA T941G and COMT Val158Met in interaction with childhood abuse and neglect.

Predictors	ΔR^2^	β	*p*	*p*(*i*)
Model 1 (Abuse)				
Step 1: Age	0.18	−0.03	0.574	0.050
Depressive symptoms		0.17	**0.010**	0.022
Step 2: Abuse	0.19	0.11	0.383	0.044
MAOA		0.08	0.346	0.033
COMT		0.08	0.377	0.039
Step 3: Abuse × MAOA	0.02	0.35	**0.003**	0.017
Abuse × COMT		0.42	**<0.001**	0.006
MAOA × COMT		−0.10	0.331	0.028
Step 4: Abuse × MAOA × COMT	0.03	−0.34	**0.002**	0.011
Model 2 (Neglect)				
Step 1: Age	0.18	−0.01	0.936	0.050
Depressive symptoms		0.43	**<0.001**	0.006
Step 2: Neglect	0.00	−0.13	0.302	0.033
MAOA		0.12	0.230	0.022
COMT		0.05	0.635	0.044
Step 3: Neglect × MAOA	0.01	0.21	0.103	0.011
Neglect × COMT		0.08	0.554	0.039
MAOA × COMT		−0.14	0.264	0.028
Step 4: Neglect × MAOA × COMT	0.01	−0.18	0.182	0.017

Note: NSSI = nonsuicidal self-injury, β = standardized beta estimate, *p*(*i*) = corrected significance level calculated using the B&H procedure [51]; if the *p*-value is less than or equal to the corresponding *p*(*i*), the result is significant (shown in bold). All continuous values were mean-centered.

## Data Availability

The data presented in this study are available on request from the corresponding author. The data are not publicly available due to privacy.

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
