# Peer review of "The Effects of Childhood Maltreatment on Non-Suicidal Self-Injury in Male Adolescents: The Moderating Roles of the Monoamine Oxidase A (MAOA) Gene and the Catechol-O-Methyltransferase (COMT) Gene"

_ijerph, 2021, doi:10.3390/ijerph18052598_

Round 1

Reviewer 1 Report

The authors of the manuscript ‘The effects of childhood maltreatment on non-suicidal self-in- jury in male adolescents: the moderating roles of the monoamine oxidase A (MAOA) gene and the catechol-O-methyltransferase (COMT) gene‘ present novel and interesting findings regarding the interaction of childhood trauma with COMT and MAOA gene polymorphisms. However, methodological issues and the presentation decrease enthusiasm for the current version.

The Introduction section was excessively long, as well as sections 2.2.1 and 2.2.2, with excessive detail, making it hard to read at times

No inclusion/exclusion criteria were presented.

It is not clear why the girls were sample if they were later excluded.

The authors report they summed the scores of individual scales for total abuse and neglect scores, but did not explain the rationale to do so, instead of examining individual dimensions of childhood trauma reported in the CTQ included in the analysis. This is relevant since certain types of trauma, e.g. sexual abuse, seem to have a particular stronger impact in the development of psychopathology. Why not do a subanalysis with the sexual abuse?

The authors should report the frequency of the different types of childhood trauma in the sample.

There was no power analyisis.

There were significant correlations between depression and abuse, neglect, and NSSI. Could you please comment on those associations in the discussion? What does a mean of 38.71 indicate on the CES-DC? For instance, is a level of above 20 or 30, mild, moderate, or severe depression?

It was good to include a homogenous sample (i.e. low income families). Although low socioeconomic status (SES) was not a variable investigated in the study, how may low SES affect childhood stress and possibly affect your model?

What are the clinical implications in the study? For example, with the knowledge of a susceptible gene and a difficult childhood (trauma, hardships), what does this mean to treat or prevent NSSI behaviors?

Minor

The statement ‘Research with community samples shows that the prevalence of adolescent NSSI ranged from 17.2% - 38.6% across regions [3,4].’ sounds somewhat odd. Regions across the world or of a particular country or continent?

Next sentence, individuals are at greater risk of suicide, not ‘suicidality’

In the sentence: ‘A broad literature has investigated the moderating role of MAOA gene polymorphism in people’s response to childhood maltreatment, albeit the inconsistent results.’ Seems like ‘the’ should by replaced by ‘with’

Reviewer 2 Report

Interesting and important scientific problem and interesting approach to mechanistic understanding.  Good to see the good understanding of statistical finesse with respect to complex problems of this type.

Please consider the following comments/suggestions in revision:

1. Abstract – why measure the specific genes you are measuring?

2. "all of the male adolescents with other combined genotypes displayed higher NSSI scores when exposed to more child abuse."  This sentence and in the abstract and the concept in the text require clarification.  It could be interpreted as 1) higher levels of NSSI are associated with higher exposures of abuse or 2) that higher levels of NSSI occurred only when the children were give a new abuse provocation.

3. Introduction – It seems appropriate to assume that brain functions underpinning emotion regulation or impulsiveness are associated with NSSI. Therefore, dopaminergic genes may be fruitful candidates for NSSI vulnerability.  This sentence needs to be modified: substitute ‘hypothesize’ for ‘assume’. 

4. The present study aimed to explore whether and how the MAOA gene, COMT gene, and childhood maltreatment interact to influence NSSI behaviors   I do not believe this was the aim – please state it more accurately if that is the case – i.e., that you wanted to evaluate the possibility of individual gene contributions and their interactions.

5. Figure 1 – please re-think this figure.  It is over-simplistic, which is okay, but it neglects, in my view, at a few possibly important inputs:  1) NSSI can influence childhood treatment directly by impacting caregiver responses to these behaviors and 2) NSSI is likely to also directly impact dopaminergic function in feedback and 3) childhood treatment also would have a direct impact on dopaminergic function.  Please think about this in redrawing the figure.  I know you want to be simplistic to the goal of the research but .  . .

6. Figure 2 – please provide non-blurry version.

7. Make sure to qualify that sample population was Chinese and the relevance of this for generalization to the population as a whole.

8. The genes measured are a subset of dopaminergic genes.  Please ensure discussion of this point.

9. Be sure to discuss that the data are retrospectively attained and therefore subject to potential error / bias of memory, etc.

10. Be sure to incorporate into the Discussion, if you can, a more dynamic view of the complex interactions of the genes and behavior(s) as I have commented already in the comments on Figure 1 above.
